# Determination of Friction Performance of High Friction Surface Treatment Based on Alternative Macrotexture Metric

**DOI:** 10.3390/ma14226895

**Published:** 2021-11-15

**Authors:** Hua Zhao, Fulu Wei, Ce Wang, Shuo Li, Jie Shan

**Affiliations:** 1School of Civil Engineering and Architecture, Nanchang University, Nanchang 330031, China; zhaohua@ncu.edu.cn; 2School of Construction Management Technology, Purdue University, West Lafayette, IN 47907, USA; 3Engineering Research Institute, China Construction Eighth Engineering Division Co., Ltd., Shanghai 200122, China; flwei@cscec.com; 4Lyles School of Civil Engineering, Purdue University, West Lafayette, IN 47907, USA; wang4101@purdue.edu (C.W.); jshan@purdue.edu (J.S.); 5Indiana Department of Transportation, Division of Research, West Lafayette, IN 47906, USA

**Keywords:** high friction surface treatment, performance metric, macrotexture, mean profile depth

## Abstract

Surface friction is currently the most common metric for evaluating the performance of high friction surface treatment (HFST). However, friction test methods such as the locked wheel skid tester (LWST) commonly provide a spot measurement. Large variations may arise in the LWST testing on curves. Based on 21 actual HFST projects, a study was performed to use a macrotexture metric, i.e., the mean profile depth (MPD) to evaluate HFST’s performance and improve its quality control (QC)/quality assurance (QA) procedures. The material properties were presented to understand the aspects of HFST. The method for calculating MPD was modified to account for the variations of macrotexture measurements. A vehicle-based test system was utilized to measure MPD periodically over an 18-month period since HFST installation. Statistical analysis was performed on the MPD measurements to identify the effects of influencing factors. Compared with the friction from LWST, MPD was equally effective in evaluating HFST performance. However, the use of MPD eliminated the errors as arisen in LWST testing and made it possible to detect surface distresses, including aggregate loss, delamination, and cracking. The expected overall MPD may be calculated by combining the MPD measurements made three months after installation at different HFST sites and used as a metric for evaluating HFST performance and QC/QA.

## 1. Background

Pavement friction plays a critical role in reducing vehicle skid crashes, especially when pavement surface is wet. So far, the conventional friction treatments such as chip seal, microsurfacing, ultrathin bonded wearing course (UBWC), and diamond grinding have been widely utilized to restore pavement friction [1]. The advantages associated with the conventional friction treatments include low initial costs, relatively lenient requirements for construction temperature, and compatible materials with existing pavements. Nevertheless, the conventional friction treatments are not capable of providing long excellent, durable surface friction performance. At friction-prone locations such as horizontal curves, interchange ramps and intersections, however, pavement surface is likely to become polished more rapidly due to frequent braking of vehicles, and friction demands may exceed the friction capability of conventional friction treatments. Over the past decade, high friction surface treatment (HFST) has been promoted as a cost-effective solution to the extreme friction demands in the United Sates [2,3]. HFST consists of a thin layer of polymer binder topped with polish-resistant aggregate and can be installed by either mechanical or manual methods. The polymer binders commonly used in HFST are epoxy resin binders specially engineered for use in bonding skid resistant materials to roadway surface. The aggregate is typically calcined bauxite with an aluminum oxide (Al_2_O_3_) content of 87% [4].

In 2018, the HFST initiative was launched by the Indiana Department of Transportation (INDOT) to enhance traffic safety at horizontal curves on two-lane rural highways. A stand-alone HFST contract consisting of 22 HFST projects with a total area of 42,783 m^2^ was let under the work type of traffic [5]. The contract was awarded to a contractor at a unit price of $19.26/m^2^ by a semi-automated method of HFST installation [6]. Eventually, 21 HFSTs were installed because the existing pavement was replaced at one site prior to the installation of HFST. Compared with the HFST installation prices reported elsewhere [7], the prices for the HFST projects of INDOT decreased noticeably. It should be pointed out that HFST’s prices are still much higher than the prices of installing the conventional pavement friction treatments [8]. Therefore, care must be taken to ensure that HFST is capable of providing durable and excellent friction performance.

The frictional performance of HFST surface is widely evaluated in terms of friction measurements by state departments of transportation (DOTs) [9,10]. One advantage of measuring friction is that surface friction represents the true frictional interaction between tire and pavement. However, there exist limitations to the use of friction measurements. First, friction can be measured in accordance with the ASTM E274, E1911, or E303 test method [11,12,13]. The friction measured by a method does not necessarily correlate directly with those measured by other methods. Second, it was reported that the ASTM E274 test method using the locked wheel skid tester (LWST) is widely used for field friction testing by state DOTs [14]. However, the LWST test is designated for measuring friction over straight, flat pavement surface. When it is used on pavement sections with complex geometry, especially horizontal curves, great errors may arise due to the nature of vehicle dynamics. Third, the LWST test produces the so-called skid number (SN) or friction number (FN) that can be measured using the standard rib or smooth tire [15,16]. Nevertheless, there is no consistent relationship between the SN measurements using these two tires [17]. This is probably why the requirements for HFST vary significantly from state to state and why the minimum friction requirement (i.e., 65 for the standard rib tire at 40 mph) [4,9] is much lower than the field test results reported elsewhere [18]. Fourth, the LWST can only measure the friction in one wheel track that may not fully represent the friction performance of the pavement.

Pavement friction depends to a great extent on the texture of pavement surface, including macrotexture and microtexture [19]. Increasingly, state DOTs are aware of the role of these two types of textures in providing friction [20,21,22,23]. Nevertheless, only macrotexture has been utilized because no device is now commercially available for measuring microtexture [14,24]. To characterize macrotexture, the mean profile depth (MPD) [25], i.e., the depth parameter of macrotexture, is widely utilized by state DOTs. MPD is dependent solely on the characteristics of macrotexture profile. There should be only one true value of MPD for a specific pavement, regardless of test method. In addition, MPD measurements can be readily made continuously in both wheel tracks at highway speeds. It is also worth noting that the calcined bauxite aggregate with an Al_2_O_3_ content of 87% or higher has a unique feature, that is, its physical and mechanical properties are dominantly determined by the content of Al_2_O_3_ [26]. This implies that both friction and macrotexture metrics, i.e., SN and MPD, are equivalent if they are used to evaluate the surface friction performance of HFST.

Presented in this paper is an effort made to evaluate the friction performance of HFST with respect to surface macrotexture and determine the macrotexture metric, i.e., MPD, based on the macrotexture data collected at 21 HFST sites in Indiana. To the authors’ knowledge, MPD can be utilized not only to evaluate the friction performance of HFST, but also to enhance the quality control (QC)/quality assurance (QA) procedures for HFST installation and identify the exact locations of HFST surface distresses such as aggregate loss, delamination, and cracking.

The next section presents the business case for the proposed research study. Following this, the technical rationale, work plan, list of deliverables, schedule, and cost estimate are presented. The qualifications of the research team are then provided.

## 2. Materials and Installation

### 2.1. Materials

The epoxy resin binder used in the HFST projects was a two-component epoxy resin binder system of ASTM C881 Type III [27] was claimed as a moisture tolerant adhesive with low modulus, high strength, and great resistance to weathering. The mix ratio of the two components was 1:1 by volume. The mixing temperature was approximately 15 °C~95 °C as recommended by the vendor. Presented in Table 1 are the main physical properties of the epoxy resin binder. The cure time was 2.5 h~6 h, depending on the temperature. The compressive strength was measured in accordance with ASTM D695 that is designated for rigid plastics with cylinder test specimens of ½″ (12.7 mm) in diameter by 1″ (25.4 mm) in height. Nevertheless, the revised AASHTO requirement for the compressive strength is measured in accordance with ASTM C579 designated for chemical-resistant mortars, grouts, monolithic surfacings, and polymer concretes, and the latter is designated for rigid plastics with right cylinder test specimens of 1″ (25.4 mm) in diameter by 1″ (25.4 mm) in height. As with the epoxy–bauxite mortar, its Poisson’s ratio was 0.29 and its coefficient of thermal expansion (CTE) was 37.72 × 10^−6^/°C [28].

Presented in Table 2 are the main properties of the calcined bauxite aggregate used in the HFST projects. The detailed information on the aggregate properties can be found elsewhere [36].

The Al_2_O_3_ content is slightly less than 87%. There are many factors such as raw bauxite, calcination operation condition, type of fuel, flame temperature, and feed amount and speed that may affect the chemical composition of calcined bauxite. Therefore, the oxide proportions may vary from plant to plant, batch to batch, and even vary across a single batch. The Deval abrasion test utilizes saturated aggregate samples and therefore can better reflect the effects of the environment, especially the durability of aggregate properties. The resistance of aggregate to polishing action of vehicle tires was measured in terms of PV-10, i.e., the polish value after 10 h polishing in accordance with the ASTM D3319 method, instead of the polished stone value (PSV) measured in accordance with the BS EN 1097-8 method in European countries [40]. It should be emphasized that the PV-10 and PSV test methods differ in many aspects, such as polishing load and time, abrasion condition, and reading scale. In particular, PV-10 is commonly measured on the main scale and PSV is measured on the F-scale.

### 2.2. Installation

Summarized in Table 3 is the general information on all 21 HFST projects, including road, traffic, and HFST installation such as type of existing pavement, annual average daily traffic (AADT), truck percentage (%), preparation of existing pavement surface, temperature, and application rates of materials. Because the installation time could last for several hours for a specific project, the temperatures indicated the daily lowest and highest temperatures. There are several important points that need to be highlighted about the installation of these HFST projects. First, the early distresses of HFST were impacted dramatically by the condition of existing pavement. The most common type of existing pavement for HFST was chip seal in Indiana. Although concerns arose about the possible impact of chip seal on the durability of HFST, it was observed that chip seal in good condition would not affect the durability of HFST in terms of interface bonding strength. Second, vacuum sweeping was an effective method for preparing the surface of chip seal. Scarification milling did not necessarily provide better interface bonding between HFST and the underlying chip seal. Third, although HFST could be installed within the range of temperatures recommended by vendors, curing an epoxy binder system at low temperatures would increase not only the cost for traffic control but also the variation in the epoxy binder system. It was demonstrated that installing HFST at higher temperatures was beneficial for HFST’s durability.

## 3. Methods and Field Testing

### 3.1. Surface Texture

Pavement surface texture indicates the deviations of a pavement surface from a true planar surface. Permanent International Association of Road Congress (PIARC) carried out a study to investigate the characteristics of pavement surface friction and texture [19]. The texture of pavement surface that affects pavement friction consists of both microtexture and macrotexture as follows:

Microtexsture: Wavelength < 0.5 mm, and

Macrotexture: Wavelength = 0.5 mm to 50 mm

The peak-to-peak amplitude varies between 0.01 and 50 mm for macrotexture profiles, and between 0.001 and 0.5 mm for microtexture profiles. It was reported that fundamentally, the friction force consists of adhesion force and hysteresis force [41]. The former depends mainly on microtexture and the latter on macrotexture. Moreover, the macrotexture dominantly affects pavement surface drainage and therefore the skidding, and water splash and spray. Currently, there is no device that is commercially available for measuring microtexture directly. Macrotexture, however, can be readily measured by a conventional method using a volumetric technique [42] or a non-contact method using a laser-based technique [43]. The characteristics of macrotexture profile are presently defined by a single texture depth parameter, i.e., MPD as follows [25]:(1)$$MPD=\frac{1}{N}\overset{N}{\underset{i=1}{\sum }}MSD_{i}$$
where *N* is the number of 100-mm long segments in an entire test section, and *MSD_i_* is the mean segment depth (MSD) of macrotexture profile in the *i*th 100-mm long segment.

### 3.2. Field Testing

A vehicle-based test system was mainly utilized for collecting the MPD data at all 21 HFST sites. This test system, consisting of two high-speed 100 kHz texture point-lasers, was originally developed for the QA of chip seal operations using the MPD measurements made in both the left and right wheel tracks at highway speeds [44]. This system was validated by the verification tests on both the INDOT friction test tracks and actual chip seal pavements. A comprehensive comparison of the MPD measurements made using this test system and a portable three-dimensional (3D) 1 kHz laser texture scanner (LTS) indicated that this vehicle-based test system is capable of measuring macrotexture accurately and efficiently. The 3D LTS was also used to make MPD measurements for new HFST surfaces. It was concluded that use of two point-lasers, one for each wheel track, is needed and anticipated to acquire the necessary information for evaluating the characteristics of texture profile. Presented in Figure 1 are the summary statistics of the texture measurements, including MPD and standard deviation (SD), made along both wheel tracks in both directions at four HFST sites within approximately one month following HFST’s installation. It is shown that random differences exist between the texture measurements in the left and right wheel tracks. However, the variations are quite consistent, regardless of test direction and HFST site.

## 4. Data Processing

### 4.1. Data Pre-Processing

As shown in Equation (1), MPD is the reduction of thousands of MSD measurements in an entire pavement test section. To calculate the MSD of macrotexture profile over each segment, both low and high pass filter cutoffs were applied to remove the wavelengths out of the wavelength range of macrotexture in the raw texture data [45]. In addition, a three-point algorithm was further applied to remove possible outliers, drop-offs, and spikes in the texture data. The pre-processed texture profile data wat then utilized to calculate MSD. Presented in Figure 2 are the typical MSD measurements made over a 15 m-long HFST surface. There are approximately a total of 150 MSD measurements in each wheel track. The average of the MSD measurements is 1.351 mm in the left wheel track and 1.154 mm in the right wheel track. Taking the average of these two MSD averages yields the MPD for this 15 m-long HFST, i.e., MPD = 1.253 mm. It should be pointed out that due to the repeated applications of moving vehicle tires, the surface in the wheel tracks tend to be polished more rapidly and prone to aggregate loss. Continuously measuring the texture information in both wheel tracks ensures a high probability of detecting early defects such as aggregate loss, delamination, and reflective cracking.

### 4.2. Effect of Distress Spikes

As mentioned earlier, two texture profiles are simultaneously measured in the left and right wheel tracks, respectively, during testing. Notice that the texture profiles are commonly generated after data pre-processing. Plotted in Figure 3, respectively, are the MSD measurements along the two left wheel tracks at the HFST site on SR-43. It is shown that the MSD measurements in both wheel tracks fluctuate around 1.0 mm. Evidently, some spikes up to 23.5 mm still remain in the texture data along the right wheel track. Because of the sensitivity of the summary statistics to spikes, it is of importance to understand those spikes and ensure that the descriptive statistics of the MSD measurement best represent the true surface texture. Nevertheless, the spikes in the collected texture data arise due to not only the laser system itself, but also the presence of surface distresses or defects, especially delamination, cracking, and localized pop-off in the HFST surface. Removing all spikes, including those due to surface distresses and defects, however, may compromise the effectiveness of the use of macrotexture metrics for QC/QA of HFST.

As illustrated earlier, macrotexture is categorized as the texture with wavelengths ranging from 0.5 mm to 50 mm and peak to peak amplitudes ranging from 0.01 mm to 20 mm. To further examine the spikes retained in the texture data, the MSD measurements and surface conditions were assessed after 7 months of service at two HFST sites, including US-24b, and SR-23, as shown in Figure 4. At the HFST site on US-24b, the spikes represent mainly the delamination in the surface of HFST. At the HFST site on SR-23, the spikes are due mainly to the surface defects, such as wide cracks and localized pop-off due to severe deterioration or raveling of cracks. Presented in Table 4 are the summary statistics of the MSD measurements before and after removing the spikes in the measurements. Evidently, the spikes affect the standard deviation and coefficient of variation more significantly than the mean (i.e., MPD). As a general rule of thumb, therefore, spikes greater than 20 mm, i.e., the maximum amplitude of macrotexture defined by PIARC, are removed from the MSD measurements.

## 5. Results and Discussion

### 5.1. Variations of MPD Measurements

Presented in Figure 5 are the MPDs and standard deviations measured periodically in both wheel tracks and both directions at all 21 HFST sites over an 18-month period after installation. As mentioned earlier, all MPD values were made by the use of the vehicle-based test system, except those over new HFSTs, i.e., at age = 0, which were measured using a 3D LTS. As shown in Figure 5, the standard deviations range between 0.08 mm and 0.45 mm with approximately 90% falling within 0.19 mm to 0.29 mm, regardless of age. For new HFSTs, their standard deviations were commonly around 0.20 mm. As age increased, the standard deviations might increase or decrease, depending on the condition of existing pavement, method of surface preparation, age, and defects in HFST surface. For an HFST after around 6 months of service, a standard deviation of 0.24 mm or larger may indicate the presence of distresses such as cracking, delamination, and localized pop-off in its surface. The MPD values range between 0.95 mm and 2.04 mm. The former was measured over an HFST after 620 days in service, and the latter occurred over a new HFST. The MPD values decreased dramatically in the first three months. Afterwards, the MPD values gradually approached 1.0 mm over time. This agrees very well with the finding reported elsewhere [8], which states that the MPD of HFST will decrease noticeably in the first three months and remain stable afterward. The relationships between MPD and the four independent variables, including AADT, Truck (volume), Age (of HFST), and Radius (of curve) were further examined by the Pearson and Spearman rank correlation methods, respectively, and the results are presented in Table 5.

As shown in Table 5, all correlation coefficients are negative, regardless of the analysis method, which indicates that MPD decreases as these independent variables increase. MPD and AADT have a weak linear relationship (i.e., Pearson correlation coefficient = −0.280), but a moderate non-linear relationship (i.e., Spearman rank correlation coefficient = −0.410). Both the Pearson and Spearman rank correlation coefficients indicate a weak relationship between MPD and Truck. The relationship between MPD and Radius is very weak based on both the Pearson and Spearman rank correlation coefficients, while there is no conclusive evidence about the significance of the linear relationship (Pearson correlation). Nevertheless, both the absolute values of Pearson and Spearman rank correlation coefficients between MPD and Age are greater than 0.30 with a p-value less than 0.05. In other words, there is a moderate relationship between MPD and Age that is statistically significant. It should be pointed out that the age of HFST is essentially a measurement of traffic applications. In short, it is possible to determine the expected MPD by combining the MPD measurements at different HFST in terms of the age.

### 5.2. Expected MPD Value

Assume there are a total of *n* HFST sites. The overall MPD in terms of all n HFST sites can be calculated as follows:(2)$$M=\frac{1}{n}\sum_{i=1}^{n}m_{i}$$
where *M* = overall MPD; *n* = number of HFST sites; and *m_i_* = MPD of the *i*th HFST site.

It is indicated that in Equation (1), there is a positive linear relationship between the overall MPD and the MPD for each HFST project. Let *m*_1_, *m*_2_, …, *m_n_* be independent random variables. Then, taking respectively the expectation and variance of both sides of Equation (3) yields:(3)$$E\left(M\right)=\frac{1}{n}\overset{n}{\underset{i=1}{\sum }}E\left(m_{i}\right)=\frac{1}{n}\overset{n}{\underset{i=1}{\sum }}MPD_{i}$$
(4)$$V(M)=\frac{1}{n}\sum_{i=1}^{n}V(m_{i})=\frac{1}{n}\sum_{i=1}^{n}SD_{i}^{2}$$
in which *E*(*M*) =expected value of *M*; *E*(*m_i_*) = expected value of the MPD at the *i*th HFST site; *MPD_i_* = expected MPD at the *i*th HFST site; *V*(*M*) = variance of *M*; *V*(*m_i_*) = variance of mi; and *SD_i_* = standard deviation of *m_i_*.

As shown in Figure 5, multiple texture tests were performed periodically over a period of 18 months, especially 0 (i.e., right after), 6, 12, and 18 months after installation. Let all the MPD measurements be independent random variables. Substituting all of the statistics of MPD measurements, i.e., MPD_i_ and SD_i_ plotted in Figure 5, into Equations (3) and (4) yields the expected values and standard deviations of the overall MPD at different ages as shown in Table 6. The expected overall MPD for new HFST is approximately 1.9 mm. It decreases to around 1.40 mm after 1 month and to 1.20 mm after 2 months. As time goes on, the MPD approaches around 1.10 mm. The standard deviations at the ages of 0 month (i.e., new), 1 month, and two months are much greater than those at the ages of 6, 12, and 18 months. The COVs are all less than 3.0% and fluctuate over time after three months. This confirms again that HFST surface may experience the greatest variability in the first three months. Therefore, the MPD measured at the age of three months should be used as a metric for evaluating HFST performance and QC/QA.

## 6. Conclusions

Great efforts have been made to broaden the understanding of HFST surface texture, especially the gaps existing in the knowledge, effective measures, data needs, limitations of field testing, and performance requirements. For a standard HFST composed of epoxy resin binder and calcined bauxite aggregate, surface friction and macrotexture metrics such as SN (or FN) and MPD are equivalent when used to evaluate the surface friction of HFST. However, the use of macrotexture eliminates the errors such as those arising from the LWST friction testing, especially on horizontal curves. Macrotexture can be readily measured continuously at highway speeds in both wheel tracks simultaneously, instead of the LWST friction that can only be measured in one wheel track. Therefore, the macrotexture-based metric may better represent the performance of HFST. In addition, MPD, the depth parameter of HFST surface macrotexture tends to become stable after a certain amount of time in service. Therefore, the MPD of macrotexture may be used as an alternative metric to friction not only for evaluating the performance of HFST, but also for enhancing the QC/QA procedures of HFST installation.

Vehicle-based test systems with two wheel-track point-lasers would provide continuous measurements for characterizing the texture profile and detecting distresses such as aggregate loss, delamination, reflective cracking, and localized pop-off. The spikes in the texture data would affect the standard deviation more significantly than the MPD. MPD would decrease dramatically in the first three months, and gradually approach 1.0 mm afterwards. It is possible to determine the expected value of MPD by combining the MPD measurements at different HFST in terms of the age. MPD measured three months after installation should be used for the purposes of performance evaluation and QC/QA. Nevertheless, field visual inspections should also be performed right after installation in case immediate corrective actions are needed.

## Figures and Tables

**Figure 1 materials-14-06895-f001:**
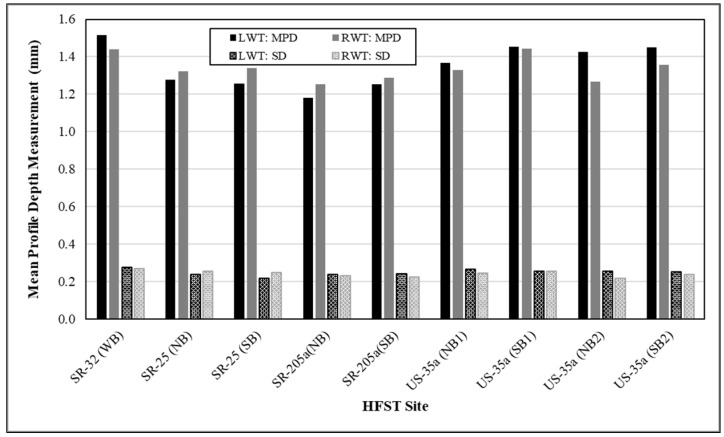
Mean profile depth (MPD) measurements in left and right wheel tracks.

**Figure 2 materials-14-06895-f002:**
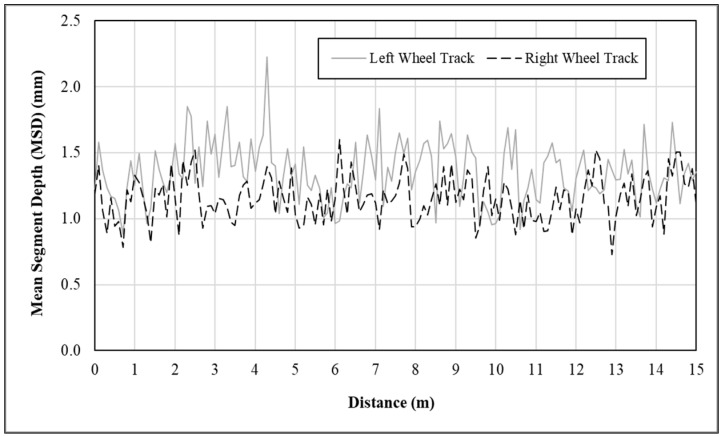
Graphical illustration of mean segment depth (MSD) measurements.

**Figure 3 materials-14-06895-f003:**
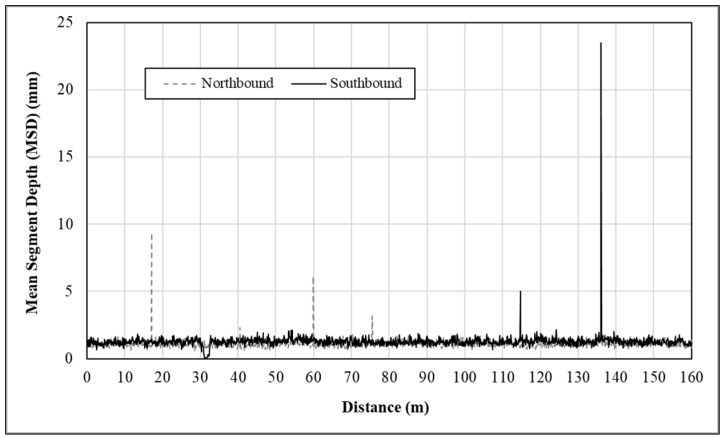
Graphical illustration of mean segment depth (MSD) measurements.

**Figure 4 materials-14-06895-f004:**
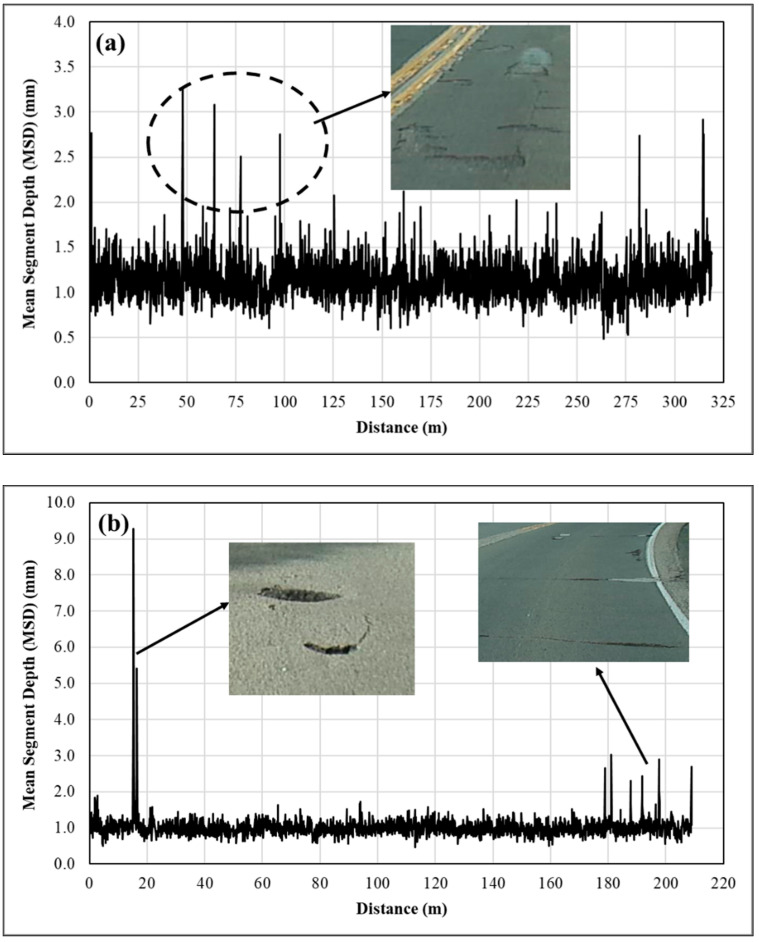
Causes for spikes in mean segment depth (MSD) measurements: (**a**) delamination, US-24b; and (**b**) pop-off and crack spalling, SR-23.

**Figure 5 materials-14-06895-f005:**
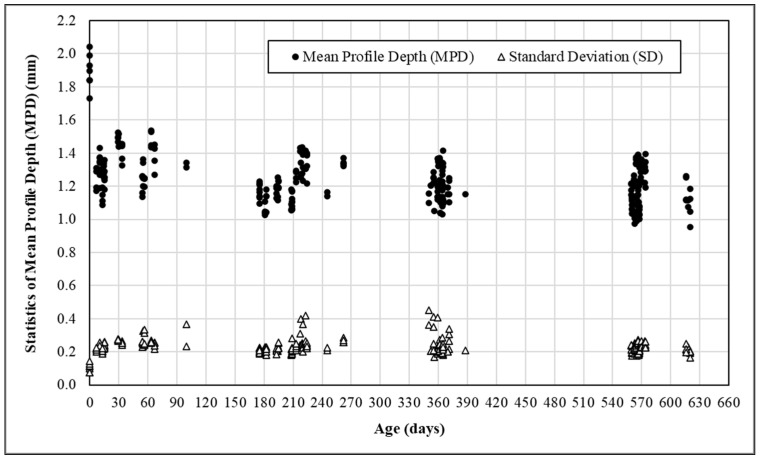
Mean profile depth (MPD) measurements over an 18-month period at 21 HFST sites.

**Table 1 materials-14-06895-t001:** Main properties of the epoxy resin binder.

Property	Test Result	Test Method
Viscosity	14.2 Poise	ASTM D2556 [29]
Gel Time	19 min	ASTM C881
Cure Rate	150 min	ASTM D1640 [30]
Ultimate Tensile Strength	19.9 MPa	ASTM D638 [31]
Elongation at Break	50.9%	ASTM D638
Durometer Hardness	65	ASTM D2240 [32]
Compressive Strength (3 h)	45.7 MPa	ASTM D695 [33]
Adhesive Strength, (24 h)	5.4 MPa	ASTM D4541 [34]
Water Absorption, (24 h)	0.11%	ASTM D570 [35]

**Table 2 materials-14-06895-t002:** Main properties of the calcined bauxite aggregate.

Property	Result	Test Method
Al_2_O_3_	86.9%	ASTM C311 [37]
Micro-Deval Abrasion	5.2%	ASTM D6928 [38]
PV-10	59.1	ASTM D3319 [39]
Aggregate Grading	Mass % Passing	
No. 6 (3.35 mm)	95–100%	95%
No. 16 (1.18 mm)	0–5%	5%

**Table 3 materials-14-06895-t003:** Information for road, traffic, pavement, and construction at HFST sites.

No.	Road	Existing Pavement	AADT	Truck (%)	Surface Preparation	Temperature (°C)	Epoxy Resin (L/m^2^) ^a^	Calcined Bauxite (kg/m^2^) ^b^
1	SR-32	New HMA	9679	5%	Shotblasting	16–29	1.95	8.31
2	US-35a	Chip Seal	2886	20%	Vacuum Sweeping	17–29	1.80	10.48
3	US-35b	Chip Seal	2886	20%		18–32	2.09	11.29
4	SR-25	Chip Seal	4282	10%		−1–11	1.83	7.70
5	SR-62a	Chip Seal	2754	7%		21–31	1.72	7.34
6	SR-62b	Chip Seal	2282	8%		21–32	1.71	6.92
7	SR-62c	Chip Seal	2176	8%		20–32	1.65	7.11
8	SR-62d	Chip Seal	2367	7%		22–32	1.81	7.03
9	SR-237	Chip Seal	685	5%		3–12	1.86	7.27
10	US-24a	Chip Seal	5205	20%	Scarification Milling	2–11	1.80	9.48
11	US-24b	Chip Seal	5205	20%		2–11	1.84	8.53
12	SR-14	Chip Seal	3675	18%		−3–11	1.97	11.30
13	SR-23	Chip Seal	5460	17%		4–11	2.07	8.46
14	SR-43	Chip Seal	2043	18%		2–14	2.31	9.17
15	SR-56	Chip Seal	133	16%		22–31	1.79	7.10
16	SR-65	Chip Seal	3207	9%		22–31	1.65	6.50
17	SR-205a	Chip Seal	3641	4%		2–16	2.07	10.44
18	SR-205b	Chip Seal	3641	4%		2–16	1.96	10.48
19	SR-257	Chip Seal	527	33%		3–13	1.75	8.12
20	SR-446	Chip Seal	1705	17%		2–14	2.00	8.92
21	SR-450	Chip Seal	888	3%		4–17	2.42	9.60

Note: ^a^ 1 ft^2^/gal = 0.0245 m^2^/liter; and ^b^ 1 lb/yd^2^= 0.543 kg/m^2^.

**Table 4 materials-14-06895-t004:** Summary statistics of MSD measurements before and after removing spikes.

HFST Site	SR-62a		US-24b		SR-23	
	Before	After	Before	After	Before	After
Mean (mm)	1.194	1.191	1.210	1.202	1.012	0.997
SD ^a^ (mm)	0.230	0.215	0.195	0.161	0.308	0.186
COV ^b^ (%)	19.3	18.1	16.1	13.4	30.5	18.7
Range (mm)	0.580–3.748		0.491–3.258		0.462–9.280	

Note: ^a^ SD = standard deviation; and ^b^ COV = coefficient of variation.

**Table 5 materials-14-06895-t005:** Summary of correlation analysis results.

(a) Pearson Correlation				
Variable	AADT	Truck	Age	Radius
Coefficient (r)	−0.280	−0.249	−0.413	−0.016
*p*-Value	0.000	0.000	0.000	0.789
**(b) Spearman rank correlation**				
**Variable**	**AADT**	**Truck**	**Age**	**Radius**
Coefficient (r)	−0.410	−0.233	−0.329	−0.176
*p*-Value	0.000	0.000	0.000	0.003

**Table 6 materials-14-06895-t006:** Expected value and standard deviation of overall MPD by age.

Age	N	EV ^a^ (mm)	SD ^b^ (mm)	COV ^c^	Lower Bound (mm)	Upper Bound (mm)
New	6	1.905	0.047	2.5	1.868	1.943
1 Month	18	1.444	0.061	4.2	1.415	1.472
2 Months	12	1.242	0.078	6.3	1.198	1.286
6 Months	72	1.222	0.028	2.3	1.215	1.228
12 Months	60	1.194	0.032	2.7	1.186	1.202
18 Months	80	1.181	0.025	2.1	1.175	1.186

Note: ^a^ EV = expected value; ^b^ SD = standard deviation; and ^c^ COV = coefficient of variation.

## Data Availability

Data will be made available based on request.

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
