# Peer review of "Determination of Friction Performance of High Friction Surface Treatment Based on Alternative Macrotexture Metric"

_materials, 2021, doi:10.3390/ma14226895_

Round 1
Reviewer 1 Report
Comments and Suggestions for Authors
Dear Authors,
- Nomenclature including list of abbreviations has to be added
- please improve the neatness of your manuscript. there are numerous typos or editing errors
- The methods or approaches used in this paper are well-known and existing, and the scientific contribution/novelty is very thin and weak. the scientific novelty of the work should be stressed out
- statistical evaluation of studies should be extended and data processing should be improved
- the conclusion should be improved, extended
Kind regards,
Reviewer
Author Response
Comment 1: Nomenclature including list of abbreviations has to be added
Response: Nomenclature has been added. Thanks for the suggestion.
Comment 2: Please improve the neatness of your manuscript. There are numerous typos or editing errors.
Response: The authors have read through the manuscript and corrected the typos and editing errors.
Comment 3: The methods or approaches used in this paper are well-known and existing, and the scientific contribution/novelty is very thin and weak. The scientific novelty of the work should be stressed out.
Response: You are absolutely right. As you know, there is a lack of well established procedures for HFST QA/QC and performance inspection in the U.S. Although the current practices by many state highway agencies are based on the friction measurements in accordance with ASTM E274 (Standard Test Method for Skid Resistance of Paved Surfaces Using a Full-Scale Tire), dramatic errors may be involved in the friction measurements due to the nature of the ASTM E274 test method. In addition, the friction measurements may vary significantly with the type of test tire. We have revised the manuscript thoroughly according to your comments. In addition, we have slightly reworded the title to better reflect the emphasis of the work.
Comment 4: Statistical evaluation of studies should be extended and data processing should be improved.
Response: We have revised the original section (i.e., Data Processing) heavily based on your comment.
Comment 5: The conclusion should be improved, extended
Response: The conclusion has received revisions.
Reviewer 2 Report
Comments and Suggestions for Authors
This is interesting research. However, the manuscript needs some modifications as follows:
- The English language should be improved throughout the manuscript. Please write the manuscript in reporting style or using passive sentences (not active sentences, e.g., we, I, the authors, etc.), see Line # 121. This comment applies to the whole manuscript.
- Lines # 157 and 158, "Error! Reference source not found..", must be deleted.
- Discussion should be extended.
- Table 4, it should be added the coefficient of variation in the table.
Author Response
Comment 1: The English language should be improved throughout the manuscript. Please write the manuscript in reporting style or using passive sentences (not active sentences, e.g., we, I, the authors, etc.), see Line # 121. This comment applies to the whole manuscript.
Response: First of all, we appreciate your encouraging comments. Respectfully, we have revised the entire manuscript thoroughly according to your comment.
Comment 2: Lines # 157 and 158, "Error! Reference source not found..", must be deleted.
Response: We have gone through the manuscript and double-checked all references and their citations.
Comment 3: Discussion should be extended.
Response: We have made major revisions to Discussion.
Comment 4: Table 4, it should be added the coefficient of variation in the table.
Response: Thanks for the suggestion. The coefficients of variation have been added. Please see Table 6 in the revised manuscript.
Reviewer 3 Report
Comments and Suggestions for Authors
This review report has been removed from the review record as it did not meet MDPI’s review report standards (https://www.mdpi.com/reviewers#_bookmark11).
Author Response
NA
Round 2
Reviewer 1 Report
Comments and Suggestions for Authors
Dear Authors,
Statistical analysis is weak. There are a number of parameters that can be used to describe surface texture. Limiting oneself to one seems to be inadequate, unreliable. Please explain your choice of parameter.
It should be considered the reasonableness of using the particular parameters presented in this paper.
You should refer to other methods currently used to indicate their strengths and weaknesses of methods, and the benefits of the method proposed in the paper, in particular showing qualitative or quantitative comparisons would be very valuable
Conclusions should be supported by the results of the paper, improved
Kind regards
Reviewer
Author Response
Comment 1: Statistical analysis is weak. There are a number of parameters that can be used to describe surface texture. Limiting oneself to one seems to be inadequate, unreliable. Please explain your choice of parameter. It should be considered the reasonableness of using the particular parameters presented in this paper. You should refer to other methods currently used to indicate their strengths and weaknesses of methods, and the benefits of the method proposed in the paper, in particular showing qualitative or quantitative comparisons would be very valuable.
Response: You are absolutely right. Surface textures are highly complex, and one texture parameter may not be enough to characterize them. As an example, three texture parameters such as height (depth), wavelength, and shape are widely used together to determine the texture characteristics of manufactured surfaces by the mechanical engineering community. However, the manuscript focuses on only one texture parameter, i.e., the mean profile depth (MPD), due to the following reasons:
- The MPD of macrotexture is the only parameter in current use by the pavement community. This is because the conventional sand patch method is currently widely used to determine MPD, especially for QC/QA of new pavements by many highway agencies. The sand patch method is also used as the only reference method for the emerging technologies such as laser-based non-contact methods. Considering the practical implementation and application, this paper does not evaluate other parameters.
- Surface texture is used as an alternative to surface friction by highway agencies. As you know, texture that affects friction includes both macrotexture and microtexture. In addition to MPD for macrotexture, we know its wavelengths are in the range of 0.5 mm to 50 mm. It was reported elsewhere [Safety Enhancement of the INDOT Network Pavement Friction Testing Program: Macrotexture and Microtexture Testing Using Laser Sensors. FHWA/IN/JTRP-2010/25. https://doi.org/10.5703/1288284314248] that the shape of macrotexture (i.e., texture slope) has weak correlation with surface friction. Instead, the shape of microtexture affects friction more than that of macrotexture.
- Nevertheless, microtexture is not in current use by highway agencies, and no devices are commercially available to fully measure pavement surface microtexture. Microtexture depends on the surface aspects of aggregate. We would also like to highlight that the standard HFST uses only calcined bauxite with a minimum Al2O3 content of 87%, and therefore the variation of aggregate surface aspects may be minimized.
Comment 2: Conclusions should be supported by the results of the paper, improved.
Response: We have reworded the conclusions to enhance accuracy.
Again, thanks for your precious time and efforts toward improving our manuscript.
Reviewer 3 Report
Comments and Suggestions for Authors
This review report has been removed from the review record as it did not meet MDPI’s review report standards (https://www.mdpi.com/reviewers#_bookmark11).
Author Response
NA
Round 3
Reviewer 1 Report
Comments and Suggestions for Authors
The evaluation of surface texture for any type of surface should be carried out comprehensively. The selection of particular parameters are related to the scale and the type of surface (application). Parameter used in the manuscript, although widely used, if analyzed separately does not give any meaningful information about the surface. A surface with different types of irregularities can have similar values of parameter evaluated in the manuscript (see Edjeou, W.; Cerezo, V.; Zahouani, H.; Salvatore, F. Multiscale analyses of pavement texture during polishing. Surf. Topogr. Metrol. Prop. 2020, 8, 024008)